# Learning to Predict Usage Options of Product Reviews with LLM-Generated Labels

## Abstract

Annotating large datasets can be challenging. However, crowd-sourcing is often expensive and can lack quality, especially for non-trivial tasks. We propose a method of using LLMs as few-shot learners for annotating data in a complex natural language task where we learn a standalone model to predict usage options for products from customer reviews. Learning a custom model offers individual control over energy efficiency and privacy measures compared to using the LLM directly for the sequence-to-sequence task. We compare this data annotation approach with other traditional methods and demonstrate how LLMs can enable considerable cost savings. We find that the quality of the resulting data exceeds the level attained by third-party vendor services and that GPT-4-generated labels even reach the level of domain experts.

## 1 Introduction

In supervised machine learning, a model's effectiveness is largely dependent on the quality of the training dataset. However, for many real-world applications the available data is unlabeled. Annotating data with meaningful labels remains one of the major challenges when developing machine learning models. Gathering labels from domain experts is time-consuming and expensive, while crowd-sourced labels are often inconsistent in quality due to varying skill levels. Although large language models (LLMs) have helped address this issue by assisting in data annotation for simpler tasks (Pangakis & Wolken, 2024; Gilardi et al., 2023; Kuzman et al., 2023), their potential remains underexplored for more intricate problems with open label sets demanding human-level language understanding. The labeled data can then be used to train a customized light-weight model on-premise, which offers higher energy efficiency, more privacy and lower long-term cost for deployment compared to a LLM. To explore the potential of automatic labeling via LLMs, we focus on the task of extracting product usage options from customer reviews. We define a *usage option* of a product as a textual phrase answering the question "What can this product be used for/as?" based on a customer review.[1]. See Figure 1 for an example.

Generating useful labels for our task presents considerable complexity, as it implicitly requires multiple decision steps: (1) Are there usage options present in the review? (2) What are the potential usage options? (3) Is the sentiment of the usage option candidates positive? Since usage options might not be unique, step (2) may require the annotation of multiple free-text labels per review, each as short as possible. While some usage options can be extracted directly from the text, others might not appear verbatim, instead requiring consideration of various contextual information. This makes it particularly challenging to label lengthy reviews correctly due to the need to consider the complete context. In addition, it can be hard to identify successful uses and exclude those where a customer's expectation was built up but ultimately disappointed. Since reviews are human-written, they contain grammatical errors and slang, degrading the readability for the annotator.

Via the task of extracting usage options from customer reviews we explore how well state-of-the-art LLMs can be utilized to label complex, real-world data. We compare estimates for training a small, specialized

---

[1]We exclude the usage option *gift* because this would apply to nearly every product. Furthermore, target audiences and product attributes are not considered to be usage options.

model with using a large, all-purpose LM in terms of energy usage and compute, and identify scenarios where the former is more efficient.

Figure 1: An example review, containing two usage options. Note that the choice of usage options is quite apparent in this case. Many other reviews are more ambiguous.

| | |
|---|---|
| **Product title:** | Pandigital Novel 7" Color Multimedia eReader -Remanufactured |
| **Review headline:** | Good Product |
| **Review body:** | This item is a nice, inexpensive way to keep in touch when away from home or another is on your main computer. It works nicely and allows you to research when hooked to a secured WiFi. It is still rather new to me and I have yet to find a good app that will allow more widespread use. I highly recommend this product. |
| **Desired Label:** | {"keeping in touch when away from home", "research"} |

## 2 Related Work

**Information Extraction from Customer Reviews**   Traditionally, information on how well a product is received used to be hand-selected from responses to customer interviews (Matzler & Hinterhuber, 1998). With the surge of online retail, large amounts of product reviews have become publicly available, leading to a rise of machine-learning-based processing (Timoshenko & Hauser, 2019; Zhang et al., 2021; Stahlmann et al., 2023). Previous work has considered the extraction of *customer needs.* Such customer needs encompass usage options, since the customer requires the product to fulfill the usage. Customer needs, however, can also be negative and include product attributes. For example "I'm getting creeped out by Google invading all aspects of our lives – I just haven't reached the point where I throw it against the wall though"[2] conveys a need, but does not contain any usage option. Timoshenko & Hauser (2019) and Zhang et al. (2021) demonstrate that the presence of customer needs can be accurately detected using LSTM networks and convolutional neural networks, while Stahlmann et al. (2023) show that Transformer networks do even better.

**Issues with Crowd-based Labeling**   A popular way to gather annotated data is to rely on crowd-sourcing platforms such as Amazon Mechanical Turk (MTurk). However, these platforms have seen a significant decline in annotation quality over the last few years (Chmielewski & Kucker, 2019). Although different strategies have been explored to train a supervised machine learning model on noisy labels from crowd-workers (Sheng et al., 2008; Ipeirotis et al., 2014), it is highly non-trivial to ensure the quality and consistency of the data and the resulting model. Manual expert labeling can attain a high quality, but is costly and time-consuming and may be impractical when large amounts of training data are required. In addition, in complex annotation tasks, even labels by experts can be noisy and inconsistent. Northcutt et al. (2021) found that for ten of the most cited datasets, an average of 3.3% of the data was mislabeled.

**LLMs for Data Labeling**   LLMs have exhibited strong zero-shot and few-shot capabilities (Brown et al., 2020) and have shown impressive performance on a variety of NLP tasks such as intent classification (Kuzman et al., 2023), quantitative reasoning tasks (Lewkowycz et al., 2022), and code generation (Le et al., 2022). Additionally, language models have been increasingly used to generate synthetic data: For data augmentation, they have been used to increase the variety of input data points in training examples with approaches such as back-translation or AugGPT (Sennrich et al., 2016; Dai et al., 2023). To increase the diversity of synthetically generated data, Chan et al. (2024) create data based on the viewpoints of 1 billion artificial personas. There have also been techniques to mitigate issues with lack of labels, for example using pseudo-labels, which are unlabeled data treated as if labeled (Lee, 2013). Another avenue is to invoke language models to provide label suggestions that may help humans with the data annotation (Desmond et al., 2021).

---

[2]Example from the dataset of Stahlmann et al. (2023) found under: `https://github.com/SvenStahlmann/HICSS-2023-Benchmarking-Machine-Learning-Models-for-Need-Identification/blob/main/data/labeled/combined.csv` (accessed July 02, 2024)

So far, labeling data with LLMs has been successfully applied to classification problems (Pangakis & Wolken, 2024; Gilardi et al., 2023; Kuzman et al., 2023) and to natural language generation (NLG) tasks such as summarization (Wang et al., 2021; Chung et al., 2022; Alizadeh et al., 2023). In this paper, we consider a more customized task, where each data point (a customer review) contains zero to multiple information sources (usage options) that need to be identified and summarized.

## 3 Methods

### 3.1 Comparing Sources of Data Annotation

To compare the quality of training labels procured from different sources, we let each labeling source annotate the same set of inputs, resulting in several training sets, which we in turn draw upon to fine-tune identical language models. To assess the quality of each labeling source, we evaluate the generalization performance of the fine-tuned models on an unseen test set. In the following, we introduce the different labeling sources.

**Experts** Experts are people deeply familiar with the task and the problem domain. In this study, the authors served as experts after spending substantial time studying and discussing the problem, the data, and devising suitable annotation guidelines. Expert labeling provides the most control over the labeling process, but is time-consuming and often costly. Labeling usage options is a mentally straining and time-consuming task, with the authors being able to label about 100 reviews per hour. The training set was divided into seven parts and annotated by seven different experts.

**Vendors** A more scalable labeling approach is the use of service vendors. These businesses have expertise in data labeling and a large workforce. We use the AWS Sagemaker GroundTruth framework, which provides a marketplace where companies can offer their labeling services. When vendors offer a contact person within the company, the risk of bad labels such as those found in (Chmielewski & Kucker, 2019) can be reduced. In addition, direct contact is useful to iteratively improve the collaboration and ensure that the workers' annotations are aligned with expectations. For this work, we chose two vendors based on prior experiences and reviews from other customers.

**LLMs** In this work, we test *GPT 3.5 Turbo* and *GPT 4* from OpenAI's GPT model family[3] as well as the largest Llama 2 model *Llama 2 70B Chat* (Touvron et al., 2023). We employ two types of prompts to generate labels for the training set: Chain-of-Thought (CoT) prompting (Wei et al., 2023) as well as a few-shot setup where we provide labeled examples in the prompt (Brown et al., 2020).

### 3.2 Evaluation

For each review, we face a set-to-set similarity problem between the usage option predictions and their references. Firstly, we view the problem as a simplified binary classification between whether a review has usage options (positive class) or not (negative class) and calculate the F1-score over all reviews. Additionally, we propose a new score suitable for this scenario: the *Multiple-Reference String Set Similarity Score* (MS4). For a set of reviews, individual scores are then aggregated using the *Harmonically-Aggregated MS4* (HAMS4). In the following, we introduce these scores in more detail.

For similarity (SIM), we use the clipped cosine similarity of the embeddings generated by the *all-mpnet-base-v2* model[4] (Song et al., 2020), which we transform twice using the cumulative distribution function of a beta distribution with parameters $\alpha \approx 1.35, \beta \approx 1.65$ and $\alpha \approx 14.72, \beta \approx 3.39$ for more evenly distributed scores. Clipping ensures similarity scores in the range of $[0, 1]$.

**MS4** Before defining MS4, we first propose a slightly simpler score, the *String Set Similarity Score* (S4). S4 is used to evaluate the similarity of two sets of strings, a prediction $\hat{\boldsymbol{u}} = \{\hat{u}_1, \ldots, \hat{u}_n\}$ and a reference $\boldsymbol{u} = \{u_1, \ldots, u_m\}$.

If $\hat{\boldsymbol{u}} \neq \emptyset$ and $\boldsymbol{u} \neq \emptyset$, S4 greedily matches each string in $\hat{\boldsymbol{u}}$ with a string in $\boldsymbol{u}$ to calculate precision $P_{S4}$, and vice versa for each string in $\boldsymbol{u}$ to $\hat{\boldsymbol{u}}$ to obtain recall. To account for semantic overlaps within $\hat{\boldsymbol{u}}$ and $\boldsymbol{u}$, we

---

[3]`https://platform.openai.com/docs/models` (accessed July 02, 2024)
[4]`https://huggingface.co/sentence-transformers/all-mpnet-base-v2` (accessed July 02, 2024)

weight a string based on the average distance to other strings in the same set. We calculate the S4 score as the harmonic mean of recall and precision.

$$P_{S4}(\hat{\boldsymbol{u}}, \boldsymbol{u}) = \frac{\sum_{i=1}^{|\hat{\boldsymbol{u}}|} w(\hat{u}_i, \hat{\boldsymbol{u}}) \cdot \max\{\text{SIM}(\hat{u}_i, u_j) \mid 1 \leq j \leq |\boldsymbol{u}|\}}{\sum_{i=1}^{|\hat{\boldsymbol{u}}|} w(\hat{u}_i, \hat{\boldsymbol{u}})}, \tag{1}$$

$$S4(\hat{\boldsymbol{u}}, \boldsymbol{u}) = \frac{2 \cdot P_{S4}(\hat{\boldsymbol{u}}, \boldsymbol{u}) \cdot P_{S4}(\boldsymbol{u}, \hat{\boldsymbol{u}})}{P_{S4}(\hat{\boldsymbol{u}}, \boldsymbol{u}) + P_{S4}(\boldsymbol{u}, \hat{\boldsymbol{u}})}, \tag{2}$$

$$w(x, \hat{\boldsymbol{u}}) = \frac{1}{|\hat{\boldsymbol{u}}|} \sum_{i=1}^{|\hat{\boldsymbol{u}}|} 1 - \text{SIM}(x, \hat{u}_i), \tag{3}$$

where $\text{SIM} \in [0, 1]$ is a similarity metric. In the cases $\hat{\boldsymbol{u}} = \emptyset$ and $\boldsymbol{u} = \emptyset$, $S4(\hat{\boldsymbol{u}}, \boldsymbol{u}) = 1$. If exactly one is empty, $S4(\hat{\boldsymbol{u}}, \boldsymbol{u}) = 0$.

Due to the inherent ambiguity of our problem, a subset of reviews in our test set contains multiple ground-truth labels. For example, even among a set of human experts, the inter-annotater agreement is mediocre and exhibits a high variance due to unclear reviews (see also Section 4.5.) In order to allow multiple possible reference sets $\boldsymbol{U} = (\boldsymbol{u}_1, \ldots, \boldsymbol{u}_k)$, we introduce MS4 as

$$MS4(\hat{\boldsymbol{u}}, \boldsymbol{U}) = \max_{\boldsymbol{u}_i \in \boldsymbol{U}} S4(\hat{\boldsymbol{u}}, \boldsymbol{u}_i). \tag{4}$$

In Appendix A.1, we also report the results on the established *Word Mover's Similarity* (WMS), which correlates with MS4.

**HAMS4** Because many reviews contain no usage options, which can only yield a score of 0.0 or 1.0, we calculate the mean performance for reviews without any usage options and reviews with usage options separately. For reviews that have bothe types of labels, we assign them to a class based on the prediction. Subsequently, we compute the harmonic mean between these two means and denote the result as the *harmonically-aggregated MS4* (HAMS4). This treats both categories of reviews as separate problems and weights them equally.

## 4 Experiments

### 4.1 Setup

**Dataset** We use the Amazon Customer Review Dataset[5] from the US marketplace, which contains around 151 million customer reviews. The original dataset has a low percentage of reviews that contain usage information. Therefore, we apply several preprocessing steps that reduce the number of reviews to ~63 million (The preprocessing steps can be found in Section A.2). We sample a total of 4,252 reviews from this dataset, which we split into three parts: a prompt selection set (252), an evaluation set (2,000), and a training set (2,000). For the training set, we use a fixed subset of 10% as a validation set. A team of experts jointly annotated the evaluation set to reduce human variance.

**Training Setup** We use the *T5* architecture (Raffel et al., 2020) for our models as it is designed for arbitrary text-to-text tasks. Specifically, we take advantage of the transfer-learning capabilities of the instruction-tuned checkpoint *Flan-T5 Base*[6] (Chung et al., 2022), allowing for a reasonable generalization performance despite our small training set.

We use Adafactor optimization (Shazeer & Stern, 2018), following prior work (Raffel et al., 2020; Chung et al., 2022) that fine-tuned *T5* models on a variety of tasks.[7] We combine this with an inverse-square root learning rate schedule with a linear warm-up, similar to what Shazeer & Stern (2018) use, originally

---

[5] https://s3.amazonaws.com/amazon-reviews-pds/readme.html (accessed November 11, 2022)

[6] https://huggingface.co/google/flan-t5-base (accessed July 02, 2024)

[7] In particular, we use the Hugging Face implementation from https://huggingface.co/docs/transformers/v4.32.0/en/main_classes/optimizer_schedules (accessed July 02, 2024) without relative step sizes suggested in their documentation.

Table 1: The hyperparameters used in training and tuned via Bayesian hyperparameter search. Fixed values are set according to empirical testing. For the number of active layers $m, n$ we try two options ($m = n = 6$ and $m = n = 12$). Additionally, we always fine-tune the language model head.

| Hyperparameter | Value | |
| --- | --- | --- |
| Number of batches to accumulate | $k$ | |
| Active encoder layers | $m$ | ▷ Train the last $m$ layers of the encoder |
| Active decoder layers | $n$ | ▷ Train the last $n$ layers of the decoder |
| Batch-size | 16 | |
| Number of epochs | $\infty$ | ▷ Early stopping with patience of five epochs |
| Learning rate factor | $\alpha^*$ | |
| Warm-up factor | 0.01 | |

Table 2: HAMS4 scores of LLM prompts on dedicated prompt selection set, best scores per model in bold.

| Model | CoT 6-shot prompt | CoT 2-shot prompt | 6-shot prompt | 2-shot prompt |
| --- | --- | --- | --- | --- |
| Llama 2 70B | 0.25 | **0.50** | 0.33 | 0.47 |
| GPT 3.5 Turbo | 0.68 | 0.74 | 0.77 | **0.79** |
| GPT 4 | 0.72 | 0.69 | **0.81** | 0.78 |

proposed by Vaswani et al. (2017). The learning rate $\alpha_t$ is therefore given by $\alpha_t = \alpha^* \cdot 10 \cdot \min(\gamma\, t, \frac{1}{\sqrt{t}})$, where $\gamma$ configures the slope of the linear warm-up and $\alpha^*$ governs the general scale of the learning rate schedule.

We keep most of the hyperparameters fixed across all training runs and report them in Table 1. To find a reasonable estimate of the optimal values for the learning rate and accumulated batch size, we use the Weights & Biases implementation of a Gaussian-Process-based sequential hyperparameter search (BHS) (Bergstra et al., 2011). As the target metric, we consider the minimum validation loss across an entire training run, which is computed after every training epoch. Each BHS consists of ten runs with early stopping and a patience of five epochs. We choose the best configuration, again, based on the validation loss. We initialize $\alpha^* \sim \text{Lognormal}(\ln 10^{-4},\ |\ln(3 \cdot 10^{-4}) - \ln(10^{-4})|)$ and $k \sim \mathcal{N}(16, 4)$, rounded to the nearest integer with min-value 1.

**Testing** To compare the performance between two models, we use permutation testing, implemented by SciPy[8]. We measure the difference in HAMS4 and sample 10,000 random permutations. We correct our $\alpha = 0.05$ with the amount of tests we apply. This results in $\alpha^* = 0.05/7 \approx 0.00714$.

### 4.2 Prompt Selection

We choose *GPT 3.5 Turbo*, *GPT 4*, and *Llama2 Chat 70B* to investigate the impact of the prompt format on label quality. All labels are generated by sampling with a temperature of 0.2. Due to resource constraints, we run *Llama Chat 70B* with float16 precision instead of float32. We compare a basic few-shot prompt to a chain-of-thought-structured few-shot prompt. Each prompt includes the same examples with a 2-shot and 6-shot variant. For the exact prompts, please see Appendix A. We measure each prompt's performance on the expert-annotated prompt selection set.

Since we evaluate results based on semantic similarity, the resulting *Llama 2* scores seem quite good (see Table 2). Unfortunately, it fails to follow our specified output format and instead describes usage options in long sentences. This makes automatic extraction infeasible and we therefore disregard *Llama 2* in the following experiments. For the GPT models, chain-of-thought prompts generally seem to reduce the quality of the labels. Based on HAMS4, we choose the 6-shot prompt for *GPT 4* and the 2-shot prompt (see Figure 6 and Figure 5) for *GPT 3.5 Turbo* to annotate the training set.

---

[8]`https://docs.scipy.org/doc/scipy/reference/generated/scipy.stats.permutation_test.html` (accessed July 02, 2024)

Table 3: Few-shot results on the evaluation set. The best scores per metric are marked in bold.

| Model | HAMS4 | Classification F1 | Mean MS4 (TP) |
|---|---|---|---|
| GPT 4 | **0.704** | **0.894** | 0.615 |
| GPT 3.5 Turbo | 0.656 | 0.833 | **0.637** |
| Flan-T5 Base | 0.023 | 0.056 | 0.392 |

### 4.3 Experts vs. Crowd Workers vs. LLMs

To compare the quality of different sources of label annotations, we fine-tune a model on the training set separately for each of the label sources (Experts, Vendor A, Vendor B, *GPT 3.5 Turbo*, and *GPT 4*). As baselines, we further assess the few-shot results of the pre-trained *T5* model without fine-tuning, as well as *GPT 3.5 Turbo* and *GPT 4*.

Table 4: Scores of the fine-tuned models on the evaluation set. The F1 classification score only considers if usage options where predicted or not and the true positive (TP) score captures the accuracy of the predicted usage options. For HAMS4, the underlined score is statistically equivalent.

| Labeling Source | HAMS4 | Classification F1 | Mean MS4 (TP) |
|---|---|---|---|
| GPT 4 | **0.618** | **0.844** | 0.542 |
| Experts | 0.598 | 0.784 | **0.617** |
| GPT 3.5 Turbo | 0.541 | 0.755 | 0.563 |
| Vendor A | 0.296 | 0.506 | 0.475 |
| Vendor B | 0.179 | 0.383 | 0.402 |

Although *Flan-T5* has achieved good few-shot results on a number of tasks (Chung et al., 2022), its performance on our task is quite poor (see Table 3). This is somewhat expected due to the task's complexity. We see a substantial increase in performance across all scores after fine-tuning. Still, none of the fine-tuned models is able to match the few-shot GPT performances directly (compare Table 4). Again, this is not unexpected as even the smaller variant *GPT 3.5 Turbo* has about three orders of magnitude more parameters than *Flan-T5 Base*. While the model trained on Vendor A's data performs significantly better than the one trained on Vendor B's data, both are significantly worse than the other labeling sources. They predict fewer usage options than the other models, which is reflected by the F1 classification score. This is likely a direct consequence of the low number of usage options in the training data. Only about 24% and 18% of labels contain usage options for Vendor A and Vendor B, respectively, while the experts find usage information in about 30% of training set reviews. Additionally, the TP score suggests a lower quality or higher variance in the predicted usage options. The *GPT 4*-trained model outperforms the one trained on *GPT 3.5 Turbo* labels, which is mirrored in their few-shot results on the evaluation set (see Table 3).

### 4.4 Feasibility

While modern LLMs offer strong few-shot capabilities, their large size and general-purpose nature lead to a high energy consumption. Finetuning a smaller model can address these limitations by focusing its training on the relevant domain and task. We measure the total number of FLOPs necessary for fine-tuning our best model as well as its average inference cost per request on the 4,000 training- and test-reviews using the PyTorch flop_counter[9]. Since both the model size of *GPT 4* and its inference costs have not been publicly disclosed by OpenAI, we work with an estimate based on the size of the largest *GPT 3* variant. We use the FLOPs-per-token estimates proposed by Kaplan et al. (2020), which give us $0.175 \times 10^{12} \cdot 2 = 0.35 \times 10^{12}$ FLOPs per token for the 175B parameter model. We scale this number by two, four and eight to get four different estimates and multiply each with the average number of generated tokens per request when labeling

---

[9]https://github.com/pytorch/pytorch/blob/f2900420da9cd96465e84071b6fe1f7c110ed527/torch/utils/flop_counter.py (accessed July 2, 2024)

Table 5: Energy consumption in FLOPs for different *GPT 4* estimates. Note that inference values are per request. *T5* training includes both the training data annotation using GPT 4 and the model training itself.

| GPT 4 inference | T5 training | T5 inference | Break-even |
|---|---|---|---|
| $1.93 \times 10^{12}$ | $17.5 \times 10^{15}$ | $0.206 \times 10^{12}$ | 10,185 requests |
| $3.85 \times 10^{12}$ | $21.4 \times 10^{15}$ | $0.206 \times 10^{12}$ | 5,862 requests |
| $7.71 \times 10^{12}$ | $29.1 \times 10^{15}$ | $0.206 \times 10^{12}$ | 3,878 requests |
| $15.4 \times 10^{12}$ | $44.5 \times 10^{15}$ | $0.206 \times 10^{12}$ | 2,927 requests |
| $3,080 \times 10^{12}$ | $6,180 \times 10^{15}$ | $0.206 \times 10^{12}$ | 2,005 requests |

with *GPT 4* (5.5065 tokens/request). The last estimate is based on unconfirmed leaks reporting *GPT 4*'s inference cost at $560 \times 10^{12}$ FLOPs[10].

Training our own T5 model incurs upfront energy costs due to both training data annotation and model training itself. However, once trained, the energy consumption for inference is a lot smaller. Even for the most conservative estimate we reach FLOPs parity around 10,000 requests, which translates to significant energy savings over extended use (see Table 5). For the largest *GPT 4* estimate, just five inference requests of the T5 model after training result in energy parity. In a realistic deployment scenario with tens of thousands of requests small, customized models become essential, as with growing inference costs, the break-even point asymptotically approaches the number of training examples labeled with *GPT 4*.

### 4.5 Error Analysis

We find that, with *GPT 4* labels, the model seems to focus more on the actions mentioned in the review, but it is often unable to distinguish which of them are the usage options of the *product*. On the contrary, the model trained with labels from *GPT 3.5 Turbo* appears to have less of an understanding of what a usage option is, which fits the improved language modeling capabilities of *GPT 4*. The predictions often include features or attributes of a product instead of applications, while at the same time explicitly mentioned usage options are overlooked. This explains the substantial difference in the classification score we observe (see Table 4). Furthermore, both mostly *extract* usage options from the review. As expected, this lack of abstraction is less present in the expert-trained model, as reflected in the TP score. However, the *GPT 4*-trained model achieves a higher classification score. This might be explained by more consistent labels, as on top of the variance between human annotators, it is hard even for the same person to remain consistent throughout the labeling process. For a set of six labels on 100 reviews, human experts achieved an inter annotator agreement (mean pairwise score) of 0.508 with a standard deviation of 0.434, while GPT-4 manages to achieve a score of 0.965 with a standard deviation of 0.107.

Overall, the results show that synthetic data for a complex task like ours can lead to better generalization than crowd-based annotations and even match the performance of expert-labeled data. The more powerful *GPT 4* model has a significantly better performance than *GPT 3.5 Turbo*; however, this improvement in accuracy comes currently with a 26-times higher cost and a longer inference time per review. Common problems with all LLMs are reviews that contain unclear opinions about the product (e.g., when a customer mentions a potential usage option that did not work out). Furthermore, reviews that are hard to understand due to missing information that must be inferred from the context, grammatical mistakes, or imprecise phrasing are problematic across all versions of our small fine-tuned model.

## 5 Limitations

First, due to our focus on usage option prediction as a single task, it is not clear to what extent all aspects of our findings generalize to other task types. However, we still believe usage option prediction to be an informative example for LLM-based labeling due to its complexity and challenging nature requiring multiple steps of reasoning.

---

[10]https://www.semianalysis.com/p/gpt-4-architecture-infrastructure (accessed July 03, 2024)

Second, we do not address mechanisms to identify hallucinated labels that LLM-based labeling can introduce to a dataset. While we qualitatively show how knowledge distillation with a smaller model can help to mitigate hallucinations (see Section A.3), we believe that an extensive analysis of their prevalence and methods to address should be studied in follow-up work.

## 6 Conclusion

In this paper, we introduced the real-world task of generating usage options from customer reviews. We showed that due to the task's complexity, the inter-annotator variance is high, even for experts, and crowd-sourced labels via professional vendor services are lacking in quality. We addressed these issues by exploring a range of LLMs for generating synthetic labels and found that labels generated with *GPT 4* were able to perform on par with expert labelers at a fraction of the cost and time. Our results further suggest that the model seems to be more consistent during the labeling process than a group of human experts. Lastly, we also illustrate the advantages of using a small, customized model which suffers less from hallucinations and is much more energy-efficient. We will make our code and the generated labels publicly available.

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

# A   Appendix

## A.1   Results with Word-Mover's Similarity

To enable better comparison with other work through a more established metric, we additionally report the *Word Mover's Similarity* (WMS) for our experiments. This similarity metric is based on the *Word Mover's Distance* (WMD) (Kusner et al., 2015), where WMS is the negative logarithm of WMD. We calculate a slightly modified version of WMD, in which we assign weights to words based on their normalized average distance to other words within the same set, instead of uniform weighting. This distance can be expressed as $1 - \text{SIM}$, where we use the same SIM as for MS4. For the cost function between two words, we opt for the negative logarithm of SIM. To ensure mathematical consistency, we set the similarity value (SIM) to $10^{-6}$ when its original value is 0.

In Table 6 and Table 7 we list the experimental results of Table 3 and Table 4 with WMS. The trends seen with HAMS4 hold true with WMS. The only difference is that the experts perform as well as GPT 4.

Table 6: Scores of the few-shot results on the evaluation set

| Model | HAMS4 | WMS |
|---|---|---|
| GPT 4 | 0.704 | 0.588 |
| GPT 3.5 Turbo | 0.656 | 0.514 |
| FLAN-T5 | 0.023 | 0.016 |

Table 7: Scores of the fine-tuned models on the evaluation set.

| Labeling Source | HAMS4 | WMS |
|---|---|---|
| GPT 4 | 0.618 | 0.533 |
| Experts | 0.598 | 0.548 |
| GPT 3.5 Turbo | 0.541 | 0.411 |
| Vendor A | 0.296 | 0.259 |
| Vendor B | 0.179 | 0.157 |

## A.2   Preprocessing

The preprocessing of the original data set was mainly the result of insights gained from self-labeling, where we found roughly 17% of the reviews to contain usage options. To label more informative samples and optimize labeling costs we increase the percentage of examples with usage options. We therefore define a set of rules on how to filter the data set:

- **Category** We filter out all media categories (e.g., (e-)books), because they are relatively large categories in which not many reviews contain a usage option. The reviews are mostly recounts of the product itself. In addition, we remove the category "gift card", because we do not view "gifting" as a usage option. Lastly, we exclude "Health and Personal care"-related reviews from our data for ethical reasons, since some claim to heal diseases with unproven methods. Overall, 46% of all reviews are removed by category-based filtering.

- **Length** We remove all reviews containing less than five words, often consisting of only the words "Good" or "Bad". We believe there is a minimum length required to define usage options. Furthermore, we cut off all reviews after 400 words, because they are hard to label for humans and our small models.

- **Customer ID** During our analysis, we identified a potential issue with bot-generated reviews. We define such a review as one which was written by a customer who wrote more than 30 reviews in one day. Approximately 8 million reviews stem from such users.

- **Verified purchases** All reviews that are neither verified nor part of the vine program are dropped, as we cannot rely on them being reputable information regarding usage descriptions.

Apart from filtering, we HTML-parse the content in the column "review_body" as it contains HTML tags which make annotation difficult.

By preprocessing the data, we reduce the number of reviews from roughly 151 million to 63 million and increase the estimated fraction of usage options. In our human-labeled evaluation dataset of 2000 reviews (see section 4.1) about 34% of the examples contain usage options.

### A.3 Hallucinations

We show three example reviews for which GPT 4 suffers from hallucinations in Figures 2-4. The desired label for all example reviews is "No usage option". The FLAN-T5 model trained on GPT 4 data (see Section 4.3) correctly predicts this answer, while using GPT 4 directly outputs hallucinated usage options. This suggests that using knowledge distillation with smaller models mitigates hallucinations. These reviews were selected to illustrate the three main types of hallucinations by GPT 4.

Figure 2: A review where GPT 4 predicts a usage option clearly refuted by the review. Note that the model based on FLAN-T5 is trained on GPT 4 data (see Section 4.3).

| Product title: | Vino's Tuscany Magnetic Wine Glass Charms, Set of 6 |
|---|---|
| Review headline: | Fell apart first use |
| Review body: | I was having a dinner party for 10 friends. I got two sets of these. They are very pretty and I was looking forward to using them. A few from each set fell apart that night at their first use. |

| Desired Label: | {"No usage option"} |
|---|---|
| GPT 4 Label: | {"dinner party"} |
| FLAN-T5 Label: | {"No usage option"} |

Figure 3: A review where GPT 4 predicts labels not fitting the definition of a usage option. Note that the model based on FLAN-T5 is trained on GPT 4 data (see Section 4.3).

| Product title: | TL Care 100% Organic Cotton Nursing Pads |
|---|---|
| Review headline: | Great deal! Soft, organic, do the job! |
| Review body: | This is an awesome deal for 3 pairs of organic cotton nursing pads. They are soft and have plenty of absorbency for my needs. They wash well, and I even put mine through the dryer. Love these! |

| Desired Label: | {"No usage option"} |
|---|---|
| GPT 4 Label: | {"absorbency for nursing needs", "washing", "drying"} |
| FLAN-T5 Label: | {"No usage option"} |

Figure 4: A review where GPT 4 predicts usage options based on the objects in the review, not the review itself. Note that the model based on FLAN-T5 is trained on GPT 4 data (see Section 4.3).

| Product title: | HP Pocket Media Drive 500 GB USB 2.0 Portable External Hard Drive . . . |
|---|---|
| Review headline: | great idea fits rite in the face of the slimline computer |
| Review body: | this is an excelent hard drive I never remove it from my hp slimline computer bay but I realy like that I could if I wanted to. |

| Desired Label: | {"No usage option"} |
|---|---|
| GPT 4 Label: | {"storage for computer data"} |
| FLAN-T5 Label: | {"No usage option"} |

### A.4 Prompt Versions

We provide examples for all four combinations of normal vs chain-of-thought prompting given two vs. six examples in Figures 5-8.

SYSTEM: You are a data labeler, tasked with extracting usage options from product reviews. I will give you a customer review for an e-commerce product. You should answer the question "What can this product be used for?" by only using information from the review author. Reply only and strictly with the list of usage options separated by a semicolon. If the review author does not mention any usage option, output "No usage options". Do not output negative usage options or further product information like product quality, attributes, target audiences, etc.
USER: This grill is perfect for home BBQs and was suprisingly also able to smoke vegetables. Apparently it can also be used for camping trips but I found it to be too big.
ASSISTANT: home BBQs; smoke vegetables
USER: This was a perfect gift for my 10 year old daughter.
ASSISTANT: No usage options
USER: [review_body]

Figure 5: Normal prompt with two examples.

SYSTEM: You are a data labeler, tasked with extracting usage options from product reviews. I will give you a customer review for an e-commerce product. You should answer the question "What can this product be used for?" by only using information from the review author. Reply only and strictly with the list of usage options separated by a semicolon. If the review author does not mention any usage option, output "No usage options". Do not output negative usage options or further product information like product quality, attributes, target audiences, etc.
USER: This grill is perfect for home BBQs and was suprisingly also able to smoke vegetables. Apparently it can also be used for camping trips but I found it to be too big.
ASSISTANT: home BBQs; smoke vegetables
USER: This was a perfect gift for my 10 year old daughter.
ASSISTANT: No usage options
USER: Love love love them they offer such storage for lip sticks and concealers. And they make your make up counter look very well kept
ASSISTANT: storage for lip sticks; storage for concealers; organize make up counter
USER: Very pretty and feminine. This blouse is made of sort of voile, so a camisole under it will be needed, but it is lovely by design and everything else. Very nice!
ASSISTANT: No usage options
USER: helps alleviate pressure from pregnancy. I like its squishyness! :)
ASSISTANT: alleviate pressure from pregnancy
USER: The printer arrived on time and was easy to set up. It prints very fast and the quality is great. It is perfect for printing pictures. I am very happy with this purchase. Update: After two weeks the quality has dedeteriorated enormously. Don't buy this
ASSISTANT: No usage options
USER: [review_body]

Figure 6: Normal prompt with six examples.

SYSTEM: You are a data labeler, tasked with extracting usage options from product reviews. I will give you a customer review for an e-commerce product. You should answer the question "What can this product be used for?" by only using information from the review author. You should first explain your thought process step-by-step, followed by the actual result ("Result:"). Your result may only be a list of usage options separated by a semicolon. If the review author does not mention any usage option, your result should be "No usage options". Do not output negative usage options or further product information like product quality, attributes, target audiences, etc.

USER: This grill is perfect for home BBQs and was suprisingly also able to smoke vegetables. Apparently it can also be used for camping trips but I found it to be too big.

ASSISTANT: The review author first mentions that the grill is perfect for home BBQs. Then, the author mentions that they used it to smoke vegetables. Finally, the author mentions that the product can not be used for camping trips because it is too big.
Result: home BBQs; smoke vegetables

USER: This was a perfect gift for my 10 year old daughter.

ASSISTANT: The review author mentions that the product can be used as a gift for their daughter, but gifting does not count as an usage option.
Result: No usage options

USER: [review_body]

Figure 7: Chain of thought prompt with two examples.

SYSTEM: You are a data labeler, tasked with extracting usage options from product reviews. I will give you a customer review for an e-commerce product. You should answer the question "What can this product be used for?" by only using information from the review author. You should first explain your thought process step-by-step, followed by the actual result ("Result:"). Your result may only be a list of usage options separated by a semicolon. If the review author does not mention any usage option, your result should be "No usage options". Do not output negative usage options or further product information like product quality, attributes, target audiences, etc.

USER: This grill is perfect for home BBQs and was suprisingly also able to smoke vegetables. Apparently it can also be used for camping trips but I found it to be too big.

ASSISTANT: The review author first mentions that the grill is perfect for home BBQs. Then, the author mentions that they used it to smoke vegetables. Finally, the author mentions that the product can not be used for camping trips because it is too big.
Result: home BBQs; smoke vegetables

USER: This was a perfect gift for my 10 year old daughter.

ASSISTANT: The review author mentions that the product can be used as a gift for their daughter, but gifting does not count as an usage option.
Result: No usage options

USER: Love love love them they offer such storage for lip sticks and concealers. And they make your make up counter look very well kept

ASSISTANT: The review author first mentions that the product can be used for storing lipsticks and concealers. Then, the author mentions that they also use it to make their make up counter look well kept.
Result: storage for lip sticks; storage for concealers; organize make up counter

USER: Very pretty and feminine. This blouse is made of sort of voile, so a camisole under it will be needed, but it is lovely by design and everything else. Very nice!

ASSISTANT: The review author mentions that the product is pretty and voile which is a personal opinion of the product and not a usage option.
Result: No usage options

USER: helps alleviate pressure from pregnancy. I like its squishyness! :)

ASSISTANT: The review author mentions that the product was helpful in alleviating pressure from pregnancy.
Result: alleviate pressure from pregnancy

USER: The printer arrived on time and was easy to set up. It prints very fast and the quality is great. It is perfect for printing pictures. I am very happy with this purchase.
Update: After two weeks the quality has dedeteriorated enormously. Don't buy this

ASSISTANT: Initially, the review author was happy with his printer and used it for printing pictures. However, after two weeks the quality deteriorated and therfore "printing pictures" is not a valid usage options.
Result: No usage options

USER: `[review_body]`

Figure 8: Chain of thought prompt with six examples.

