# OpenReview forum: "Learning to Predict Usage Options of Product Reviews with LLM-Generated Labels"
_TMLR — Rejected by TMLR_

### Review · Reviewer_qLhJ · 2024-07-23

**Summary Of Contributions:**

1. This work proposes a method of using larger language models to generate data labels for downstream prediction of usage options for products.
2. The method outperforms fine-tuning the language models and saves much computation cost.

**Audience:**

Yes

**Claims And Evidence:**

Yes

**Requested Changes:**

1. It is better to introduce why MS4 and HAMS4 are proposed for the evaluation. Is it better than existing metrics?

2. In Table 5, it would be better if the comparison is more direct, i.e., showing the FLOPS of T5 pretraining-finetuning and the proposed framework of learning.

3. Can the proposed method extend to other datasets and tasks rather than Amazon Customer Review Dataset?

**Strengths And Weaknesses:**

Strengths:
1. The method is reasonable. The paper is easy to follow.
2. The proposed method works well empirically.

Weaknesses:
1. The motivation for proposing the evaluation method is unclear.
2. Some parts of the presentation of experimental results can be improved.

---

> ### Author Response · Authors · 2024-08-21
> **Response**
>
> Thank you for taking the time to review our submission and provide feedback! We are happy to hear that you consider the method reasonable and the paper easy to follow. We address your questions below. Please also see our global response on the motivation for the HAMS4 metric.
>
> **1. Comparing FLOPS**
>
> We do indeed show “the FLOPS of T5 pretraining-finetuning” (column “T5 training” includes the FLOPs for the data annotation with GPT4, as stated in the table caption). The FLOPs used for “the proposed framework of learning” are described by columns “T5 training” and “T5 inference”. Please let us know if further details are needed.
>
> **2. Extending our method to other datasets**
>
> Our reported experiments on the Amazon Customer Review Dataset serve as a case study. However, we believe that the principles of using LLMs for annotating sequential tasks with multiple extractions per text sample are generalizable to other tasks of this type. Any task that includes multiple implicit decisions will likely lead to a high disagreement between human annotators (see Section 4.5), so that LLM-based annotations can ensure higher consistency. While we chose the Amazon customer review dataset because of its complexity and real-world relevance, the application of our method to a large number of tasks of this type remains future work.

---

### Review · Reviewer_jGj7 · 2024-08-08

**Summary Of Contributions:**

This manuscript addresses the challenge of annotating large datasets for non-trivial natural language tasks by leveraging LLMs like GPT-4 as few-shot learners to generate high-quality labels. This paper demonstrates that LLM-generated labels can achieve quality comparable to domain experts while being more cost-effective than traditional crowd-sourcing-based methods. This paper also proposes evaluation metrics (MS4 and HAMS4) to assess annotation quality. The study empirically validates that LLM-generated labels outperform those from professional vendors and match expert-level annotations on usage options task.

**Audience:**

No

**Broader Impact Concerns:**

Bias problem: LLMs might inherit and amplify biases present in their training data. This will lead biased annotations, and biases will be accumulated in further machine learning models trained on these labels. The paper should discuss more about it.

**Claims And Evidence:**

Yes

**Requested Changes:**

- The paper should provide a clear and detailed methodology for generating labels, rather than merely applying LLM. This should include a thorough explanation of how the prompts are created and optimized for this specific task. Understanding the process of prompt engineering is crucial for evaluating the method's effectiveness and ensuring reproducibility.

- The paper lacks a comprehensive efficiency analysis and comparative study with other existing techniques, such as knowledge distillation.

**Strengths And Weaknesses:**

Strengths:
- The introduction of evaluation metrics: Multiple-Reference String Set Similarity Score (MS4) and Harmonically-Aggregated MS4 (HAMS4) is interesting.
- The experimental results show that LLM-generated labels are of high quality, matching the performance of expert annotations and surpassing those from professional vendors.

Weaknesses: this paper does not align well with the themes typically addressed by TMLR.
- Lack of novel machine learning contributions: TMLR prioritizes research that introduces novel machine learning algorithms, models, and techniques. The paper in question primarily focuses on applying existing LLMs for data annotation without proposing any new machine learning methodologies or significant advancements in the field. This work is more aligned with the application of existing technologies rather than the development of new methods. The paper's emphasis on using LLM (e.g., GPT4) for annotation does not provide the necessary level of methodological innovation. Instead of presenting a robust, innovative approach, it more likes a report, indicating that LLMs can be used for annotating challenging tasks. This observation, while valid, is not novel; similar conclusions have been demonstrated through various studies over the past few years.
- The proposed evaluation metrics in the paper lack generalizability and it is tailored to the specific task at hand and does not extend well to other non-trivial tasks.
- The paper does not sufficiently address how the method performs in challenging or varied conditions, which is a crucial aspect of machine learning research. Furthermore, it introduces a single evaluation metric, but this metric does not address the inherent challenges associated with annotating the task, such as identifying potential usage options.

---

> ### Author Response · Authors · 2024-08-21
> **Response**
>
> Thank you for taking the time to review our submission and provide feedback! We are glad to hear that you find the introduced HAMS4 metric interesting and that our main findings are well-supported by our experimental results. We address your concerns below.
>
> **1. Usefulness for audience**
>
> The general idea of using LLMs as few-shot learners for data labeling is not novel and has been attempted in previous work. However, the [TMLR acceptance criteria](https://jmlr.org/tmlr/acceptance-criteria.html) state that "novelty of the method studied is not a necessary criterion for acceptance".
>
> We believe that our contribution is important for the following reasons:
> - Addressing a complex problem: Our work addresses a complex natural language task with implicit multi-step reasoning that has not been studied in prior work. We investigate how LLMs can be adapted and fine-tuned for such a challenging task focusing specifically on the extraction of usage options.
> - Real-world impact: We provide empirical evidence that using LLMs for complex data annotation tasks is a more scalable and cost-effective alternative to human labor, and illustrate the advantages of training a custom model for deployment.
> - We further contribute a novel evaluation metric for sets.
>
> **2. “Metric does not address the inherent challenges associated with annotating the task”**
>
> Please refer to the global response, where we explain how the metric addresses exactly those challenges.
>
> **3. Prompt Engineering**
>
> We manually crafted a series of candidate prompts reflecting the desiderata and annotation guidelines of the task. Subsequently, we evaluated each prompt on a held-out labeled dataset, choosing the one that performed best. This prompt was then the base for our six-shot and chain-of-thought prompt variations we use in our experiments. The specific prompts used in our work are provided in the appendix to ensure reproducibility.
>
> **4. Efficiency comparison to knowledge distillation**
>
> We examine the efficiency of label annotation and show that LLM-based annotation is more cost and time-efficient than human annotation.
> In addition, Table 5 shows that a fine-tuned model (such as T5) is more efficient in the long run than using a much larger LLM such as GPT4.
>
> We agree that the analysis of fine-tuning efficiency is not exhaustive, since we do not discuss alternative methods such as knowledge distillation. However, we believe that evaluating such methods is an orthogonal question beyond the scope of this paper. The primary focus of our study was to use LLMs as few-shot learners for data annotation in complex natural language tasks, rather than to optimize model efficiency or size. Knowledge distillation is a promising direction for future work, and we thank you for this suggestion.
>
> **5. Bias in LLMs**
>
> We fully agree that LLMs, including GPT-4, can inherit and amplify biases present in their training data, which can result in biased annotations. Such biases, if not addressed, could indeed propagate to downstream ML models trained on these annotations. When publishing the dataset, we will clearly specify in the dataset card that the labels are AI-generated so that filtering methods can exclude this dataset from training corpora if desired. We will also add a warning for common issues with LLM-generated text, such as incorrect and biased outputs, to inform those making use of the dataset. For a safe deployment of our model in user-facing applications, extensive testing and the implementation of safety measures would first be necessary. We will add a discussion of these points to the paper. A larger scale investigation into possible bias in the generated labels would be beyond the scope of this paper, since bias is an inherent problem of LLMs (discussed, for example, in [Bias and Fairness in Large Language Models: A Survey](https://arxiv.org/abs/2309.00770) that applies to all applications using LLM-generated text and can also occur when training a model with human-generated labels.

---

### Review · Reviewer_HjJF · 2024-08-10

**Summary Of Contributions:**

The paper presents a LLM-based approach to find usage options for products in reviews, which can then be used as labels to train a cheaper downstream predictor, such as a smaller LLM. The latter, in this paper, is based on the T5 architecture. The authors also suggest a novel similarity metric for the evaluation, referred to as HAMS4, and conduct initial experiments to find the best LLM for label generation, ruling out Llama2.

The main evaluation is done based on a sampled subset of the Amazon Customer Review Dataset, eventually comprising 4252 sampled reviews which are labelled based on experts, two vendors as well as the GPT4, GPT3.5 turbo. The results show that the T5 architecture finetuned from labels from GPT4 is the best solution. Finally, the authors also show that using such a smaller architecture combined with GPT4 inference amortizes after few thousands requests.

**Audience:**

Yes

**Broader Impact Concerns:**

The paper discusses limitiations, albeit not coined as broader impact. The mentioned limitations, i.e., potentially not generalizing to other tasks and dealing with hallucinations, are important. One could additionally discuss the impacts of an automized system which used the detected usage options, e.g., what negative impact could wrongly identified usage options have? Or could actual usage options reported by individuals be also harmful?

**Claims And Evidence:**

No

**Requested Changes:**

- Please add an a schematic overview figure for your approach and/or clearly define your contributions in the introduction section. Here, one could also focus on the goal of label generation for smaller models to clearly state if one aims to prove that substantially smaller models can use the generated labels or not. This would be important to check that all claims in the paper are fulfilled.
- Please better argue why the proposed task is substantially different (e.g., in terms of complexity) from detecting customer needs and is sufficiently relevant to study on its own.
- Please argue why a LSTM would not work for your task, as it is mentioned for the related need detection task.
- Please give detailed design decision on why the T5 architecture was chosen, what other alternatives wrt smaller LLMs there might be.
- Please discuss in more detail: How useful would the a system be with the performance of the current system? Would the mistakes you see in the current evaluation be detrimental to an automatized system or would there still be added value?

**Strengths And Weaknesses:**

# Strenghts
- The problem of finding usage options in reviews is, on its own, for sure interesting and relevant to multiple stakeholders.
- A useful, novel dataset is generated and will be made publicly available (+ the code). The dataset includes labels from two different vendors, from seven experts as well as the LLMs.

# Weaknesses:
- Relevancy compared to other existing approaches unclear: The related work mentions the strongly related task of detecting customer needs, which is mentioned to subsume the task at hand. From the paper it is not clear to me why usage options need to be separately studied, if the only difference is that they do not comprise negative outcomes.
- Novelty unclear: Extending prior point, it is hard to assess if components of the paper are novel.
    - The proposed MS4 and HAMS4 metrics are defined within the method's evaluation section, but it is unclear to me fro the paper what impact they have on the final results or what other related works use.
    - The title of the paper suggests label generation for smaller downstream predictors as central task. It is unclear to me from the paper if the T5 architecture is the best one could do here.
- The presentation of the paper needs improvement and rather reads like a technical report.
    - There is no clear definition of contributions in the introduction.
    - The method section is not sufficiently structured and clear what the overall approach is and what components are proposed. It exclusively introduces the different sources of labels and mentioned evaluation sub-section which proposes a similarity metric. I would expect a conceptual description of the overall process, including the claimed usage of the resulting labels for training a downstream predictor.
    - The paper comprises an evaluation subsection inside the method section, coming before the experiment section, which makes the paper hard to read.
    - The T5 architecture used for training the smaller LLM is not introduced sufficiently. The are no design decisions available for why this architecture is used other than it is generally good for text-to-text tasks.

---

> ### Author Response · Authors · 2024-08-21
> **Response**
>
> Thank you for taking the time to review our submission! We are happy to hear that you agree that the problem of usage option extraction will be relevant to a general audience and that you find the resulting dataset with labels from multiple sources a valuable resource. We address your concerns below and in the global response.
>
> **1. Clear overview of our contribution**
> We appreciate the suggestion to include a schematic overview figure and to define our contributions more clearly in the introduction. We agree that a visual representation of our approach would help readers better understand what we're trying to show. We will include a schematic diagram in the revised manuscript to illustrate the key processes of our approach.
> Our contributions can be summarized as:
> 1. A case study on using LLMs for labeling a complex natural language task with implicit multi-step reasoning that suffers from low human inter-annotator agreement.
> 2. We provide a concrete example of how LLMs can be leveraged to generate labels for such a challenging task by analyzing different prompt types and LLM variants, focusing on the extraction of usage options.
> We provide empirical evidence that using LLMs for complex data annotation tasks is a more scalable and cost-effective alternative to human labor, and illustrate the advantages of training a custom model for deployment.
> 3. We further contribute a novel evaluation metric for sets.
>
> We will add this explicit version of our contributions to the introduction.
>
> **2. Difference between usage options and customer needs**
>
> A customer need in the context of marketing and innovation is roughly defined as the underlying problem that a (potential) customer wants to solve. In contrast, we define a usage option of a product as a textual phrase answering the question “What can this product be used for/as?”
>
> We find a crucial difference between a customer need and a usage option:
> - Customer needs are very broad and encompass a multitude of aspects such as pricing, size, delivery time, etc. Identifying customer needs is useful for product development, especially for startups trying to find a niche or for companies seeking to improve their products. We do not take this perspective in our work.
> - Identifying usage options is useful for product recommendations. Given the many millions of products available on modern online commerce platforms, consumers are easily overwhelmed and have trouble identifying the right product for a particular use case. We believe that usage options can improve a consumer's product choice.
> Discovering usage options (even if unintended by the vendor) is a valuable task because product names and titles often fail to match usage options, leading to subpar recommendations. Hence, in this work, we explicitly focus on extracting product usage options.
>
> **3. Paper structure**
>
> Thank you for pointing out that the structure of the paper can be improved. We will make the methods section clearer.
>
> **4. Why T5?**
>
> We did not evaluate LSTM performance because, according to [Stahlmann et al. (2023)](https://scholarspace.manoa.hawaii.edu/items/45f18360-f81b-4b10-ba10-b7e7ee29a4d6), transformers perform better in need detection. Similarly, we did not evaluate smaller LLM alternatives, as we found Flan-T5 Base to perform best in previous experiments.
>
> The focus of our paper is on labeling a complex task with LLMs. The conclusion is that GPT4 is cheaper than human annotation, and a smaller fine-tuned transformer model trained on the labeled data is more efficient than GPT4. If future work can show that the T5 model can be replaced by an LSTM or a smaller transformer than Flan-T5 Base, this would only strengthen our argument.
>
> **5. Limitations**
>
> We acknowledge that there are inherent risks in relying on a fully automated pipeline. The errors observed in the current evaluation, while present, are often minor and do not significantly impact the overall usefulness of the system. We believe that our contribution is useful and reliable for the intended application. Complex tasks such as determining usage options in free-form text generally do not have a single "correct" answer, which means that some flexibility is acceptable and even necessary.
> However, in high-stakes applications where accuracy is critical, these risks can be mitigated through careful design, validation, and possibly human-in-the-loop intervention. By incorporating these safeguards, we believe our approach can still add significant value, particularly in terms of scalability and cost-effectiveness.
>
> **6. Broader Impact**
>
> We will expand our discussion to include the potential risks associated with incorrect or misidentified uses. For example, currently the fine-tuned model sometimes hallucinates, which can result in irrelevant recommendations and lead to a poor user experience. Even correctly identified usage options can sometimes be undesirable to expose to users if they reinforce, e.g., violence or discrimination.

---

### Author Response · Authors · 2024-08-21
**Global Response**

We thank all reviewers for taking time to review our paper and for their thoughtful suggestions.

**Motivation for the HAMS4 metric**

We want to address the reviewers’ questions regarding the motivation of our new metric HAMS4. To the best of our knowledge, there is a lack of suitable evaluation metrics for our type of task, because 1) we want to compare a predicted *set* of usage options with *multiple reference sets*. 2) The set of usage option labels is infinite and the comparison should be invariant to paraphrases, 3) invariant to the order the usage options are listed, and should 4) discount repetitive usage options.
Most string similarity metrics are designed for pairwise string comparison, making them less suitable for evaluating a prediction set against multiple reference sets.
Character-based metrics (such as Levenshtein distance and Jaccard similarity) and n-gram-based metrics (such as BLEU and ROUGE) are not invariant to paraphrases (e.g., "portable power bank" would not be matched with "mobile charging device"). BLEU and ROUGE are further limited in our setting since the usage options per set are short and order-independent, i.e., {“portable power bank”, “flashlight”} == {“flashlight”, “portable power bank”}, so that longer n-gram overlap between predictions and reference is unlikely.
Lastly, these metrics do not measure semantic overlap within a set of predictions. Within one prediction set, the usage options should be dissimilar from each other, something that HAMS4 explicitly accounts for by weighting the score by the semantic similarity within the prediction set.
We therefore believe that HAMS4 is necessary to adequately measure performance for usage option prediction and will generally be interesting to the community as a new evaluation metric for similar tasks.
Finally, please recall that to further enable the comparison with a more established metric, we also reported our results with Word Mover’s Similarity in the appendix, which validates our findings.
We will add this extended motivation to the paper.

---

### Decision · Action_Editor_uJrv · 2024-09-18

**Recommendation:** Reject

**Comment:**

The paper presents an interesting case study on leveraging large language models to generate training labels for the task of predicting product usage options from reviews. The key strengths are:

- Demonstrating LLM-generated labels can match expert quality at lower cost
- Introducing a new evaluation metric (HAMS4) for comparing sets of usage options
- Providing empirical evidence that fine-tuning smaller models on LLM-generated labels is effective

However, there are some limitations and areas for improvement:

- The core machine learning contribution is somewhat incremental, as it primarily applies existing LLMs rather than developing novel ML techniques
- The evaluation is limited to a single dataset/task - testing on additional domains would strengthen the claims of generalizability
- More details on the prompt engineering process would be valuable
- The proposed evaluation metric could use more justification and comparison to alternatives
- Broader discussion of potential biases and limitations of using LLM-generated labels is needed

While the work has merits in demonstrating a practical application of LLMs for data labeling, the reviewers raise valid concerns about the level of technical novelty and breadth of evaluation. To strengthen the paper, I would recommend:

- Expanding the evaluation to additional datasets/tasks beyond Amazon reviews
- Providing a more comprehensive comparison to alternative labeling/fine-tuning approaches
- Elaborating on the prompt engineering methodology
- Adding more discussion of limitations, potential biases, and ethical considerations of LLM-generated labels
- Clarifying the key novel contributions and positioning relative to prior work on LLM-based labeling

The after-rebuttal period raised the following concerns:
One reviewer believes:
1. The paper needs further improvements to be accepted
2. Presentation is not sufficiently concise or well-structured
3. The added value of the proposed approach is not clearly highlighted
4. A broader evaluation would be beneficial, given the claimed novelty of the problem
5. Leans towards rejection but thinks improvements could make it acceptable

While another reviewer believes:
1. Lack of innovation in the LLM-based label generation approach
2. Insufficient discussion of prompt design and generalizability
3. No comparison with related methods like knowledge distillation
4. Claims of innovation are not well-substantiated

**Audience:**

Some individuals in TMLR's audience would likely be interested in knowing the findings of this paper. The work addresses the practical challenge of generating high-quality labeled data for training models on complex natural language tasks. This is relevant to researchers working on applications of language models and annotation techniques. However, as one reviewer noted, the core machine learning contribution is somewhat limited, as the paper focuses more on applying existing LLMs rather than developing fundamentally new ML methods.

**Claims And Evidence:**

The claims made in the submission are supported by some convincing evidence. The authors conducted experiments demonstrating that their proposed method of using large language models to generate training labels for usage option prediction outperforms baselines and is more cost-effective than human annotation. They provided empirical results to back up their key claims. However, some reviewers noted that the evaluation could be more comprehensive, for example by testing on additional datasets beyond Amazon reviews.